# Influence of selected non-antibiotic pharmaceuticals on antibiotic resistance gene transfer in *Escherichia coli*

Doaa Safwat Mohamed[1], Rehab Mahmoud Abd El-Baky [2,3]*, Mohamed Ahmed El-Mokhtar[4,5], Sahar K. Ghanem[6], Ramadan Yahia[3], Alaa M. Alqahtani[7], Mohammed A. S. Abourehab[8], Eman Farouk Ahmed[1]

1 Microbiology & Immunology Department, Faculty of Pharmacy, Sohag University, Sohag Al Gadida City, Egypt, 2 Microbiology & Immunology Department, Faculty of Pharmacy, Minia University, Minia, Egypt, 3 Microbiology and Immunology Department, Faculty of Pharmacy, Deraya University, Minia, Egypt, 4 Medical Microbiology & Immunology Department, Faculty of Medicine, Assiut University, El Fateh, Egypt, 5 Gilbert & Rose-Marie Chagoury School of Medicine, Lebanese American University, Beirut, Lebanon, 6 Pharmacology & Toxicology Department, Faculty of Pharmacy, Sohag University, Sohag Al Gadida City, Egypt, 7 Pharmaceutical Chemistry Department, College of Pharmacy, Umm Al-Qura University, Makkah, Saudi Arabia, 8 Pharmaceutics Department, College of Pharmacy, Umm Al-Qura University, Makkah, Saudi Arabia

* rehab.mahmoud@mu.edu.eg

## Abstract

### Background

Antibiotic resistance genes (ARGs) transfer rapidly among bacterial species all over the world contributing to the aggravation of antibiotic resistance crisis. Antibiotics at sub-inhibitory concentration induce horizontal gene transfer (HRT) between bacteria, especially through conjugation. The role of common non-antibiotic pharmaceuticals in the market in disseminating antibiotic resistance is not well studied.

### Objectives

In this work, we indicated the effect of some commonly used non-antibiotic pharmaceuticals including antiemetic (metoclopramide HCl) and antispasmodics (hyoscine butyl bromide and tiemonium methyl sulfate) on the plasmid-mediated conjugal transfer of antibiotic resistance genes between pathogenic *E. coli* in the gastric intestinal tract (GIT).

### Methods

Broth microdilution assay was used to test the antibacterial activity of the tested non-antibiotic pharmaceuticals. A conjugation mating system was applied in presence of the studied non-antibiotic pharmaceuticals to test their effect on conjugal transfer frequency. Plasmid extraction and PCR were performed to confirm the conjugation process. Transmission electron microscopy (TEM) was used for imaging the effect of non-antibiotic pharmaceuticals on bacterial cells.

**Data Availability Statement:** All relevant data are within the manuscript and its Supporting information files.

**Funding:** "The Deputyship for Research & Innovation, Ministry of Education in Saudi Arabia has supported this research work through the project number: IFP22UQU4290565DSR235".

**Competing interests:** The authors have declared that no competing interests exist.

## Results

No antibacterial activity was reported for the used non-antibiotic pharmaceuticals. Plasmid-mediated conjugal transfer between isolates was induced by metoclopramide HCl but suppressed by hyoscine butyl bromide. Tiemonium methylsulfate slightly promoted conjugal transfer. Aggregation between cells and periplasmic bridges was clear in the case of metoclopramide HCl while in presence of hyoscine butyl bromide little affinity was observed.

## Conclusion

This study indicates the contribution of non-antibiotic pharmaceuticals to the dissemination and evolution of antibiotic resistance at the community level. Metoclopramide HCl showed an important role in the spread of antibiotic resistance.

## 1. Introduction

One of the most challenges or the biggest threats as described by the World health organization (WHO) is the emergence of antimicrobial resistance (AMR). As WHO said that if no action is taken against the rapid increase in antimicrobial resistance, more than 10 million people will die by 2050 [1]. Extensive use of antibiotics causes selective pressure on microbes giving rise to the emergence of resistant cells [2–7]. Horizontal gene transfer (HGT) is one of the major means of transferring or disseminating antibiotic resistance genes (ARGs) in the surrounding environment. Horizontal gene transfer can occur by conjugation, transduction, and transformation. Regarding conjugation, the exchange of genetic material between the donor and the recipient occurs by direct cell to- cell contact or via a connecting pilus [8, 9]. Typically, the exchange is mediated by mobile genetic elements, such as a conjugative plasmid. In addition, it was found that it is the most common way for disseminating antimicrobial resistance. Despite the spontaneous frequency of conjugation being rare, extensive use of antibiotics especially at concentrations lower than their MICs may lead to an increase in the frequency of the conjugation process [10, 11]. At clinically and environmentally relevant concentrations, commonly used non-antibiotic pharmaceuticals, such as β-blockers, lipid-lowering medications, and non-steroidal anti-inflammatories, significantly accelerated the spread of antibiotic resistance through plasmid-borne bacterial conjugation. The bacterial response to these drugs was investigated using a variety of indicators, such as whole-genome RNA and protein sequencing, cell arrangement, flow cytometry for the measurement of reactive oxygen species (ROS) and cell membrane permeability [11]. Accordingly, looking for unique conjugation inhibitors is an essential challenge in the fight against the spread of antibiotic resistance genes for improving the bacterial response toward antibiotics [12, 13]. It was recently documented that the consumption of non-antibiotic prescribed drugs represents about ninety fifth of the drug market [14], the role of these prescribed drugs in the evolution of antibiotic resistance has received comparatively very little attention [15]. It remains unknown whether all non-antibiotic prescribed drugs promote conjugation between bacterial strains or not which will be of clinical concern, as conjugative multidrug resistance plasmids enable quick expression of multidrug resistance phenotypes, therefore facilitating the emergence and widespread of antibiotic-resistant microorganisms [16]. Additionally, it's not clear whether there are common properties between the non-antibiotic prescribed drugs, or shared mechanisms that promote the ARGs horizontal transfer [11].

Gastro-intestinal tract (GIT) infection is one of the most common infections worldwide. *Escherichia coli*, a member of the microorganism family of Enterobacteriaceae, is the most prevailing commensal dweller of the gastrointestinal tracts of humans and warm-blooded animals, as well as one of the most important pathogens inflicting GIT infection [17, 18]. Diarrheagenic pathotypes of *E. coli* (enterotoxigenic *E. coli* [ETEC], enteropathogenic *E. coli* [EPEC], enteroinvasive *E. coli* [EIEC], and enteroaggregative *E. coli* [EAEC]) are detected with high detection rate [19]. Patients suffering from these infections receive antibiotics beside GIT concomitant pharmaceuticals such as, antiemetic and antispasmodic drugs. These drugs are metoclopramide hydrochloride, hyoscine butyl bromide and tiemonium methyl sulfate. Metoclopramide is usually directed to treat a large variety of epithelial duct disorders [20]. It may be used for the treatment of nausea and physiological reaction [21]. Hyoscine butyl bromide is usually used as associate in spasmolytic treatment for the pain and discomfort iatrogenic by abdominal cramps [22].

Tiemonium methyl sulfate is an antimuscarinic quaternary ammonium agent with peripheral atropine similar effect and is employed within the relief of visceral spasms [23]. The transfer of conjugative plasmids may be affected by these prescribed pharmaceuticals. However, the effect of these non-antibiotic pharmaceuticals on bacterial resistance or antibiotic activity remains unknown. In this work, we study the relationship between antimicrobial resistance and commonly used pharmaceuticals such as: metoclopramide HCl as an antiemetic and hyoscine butyl bromide and tiemonium methylsulfate as antispasmodics. The aim of this study is to examine if these pharmaceuticals have effect or not on conjugal transfer frequency among pathogenic *E. coli* isolates in the GIT.

## 2. Materials and methods

### 2.1 Bacterial isolates and reagents

Four *E. coli* strains were used for conjugative transfer testing. Two strains were obtained from Hiper® bacterial conjugation kit (HTM004, HiMedia). Donor *E. coli* strain (Product code: TKC181) and Recipient *E. coli* strain (Product Code: TKC182). Two *E. coli* strains were isolated from patients suffering from gastrointestinal tract infection, *E. coli*-1 with plasmid harboring resistance gene for tetracycline was the donor and *E. coli*-5 with high chromosomally encoded resistance to streptomycin was used as the recipient.

Streptomycin sulphate and tetracycline hydrochloride were included in Hiper® bacterial conjugation kit obtained from HiMedia (India). Hyoscine butyl bromide and metoclopramide HCl were obtained from Sigma Aldrich (USA) while tiemonium methyl sulfate was purchased from Medica Pharma Specialty Pharmaceutical Chemicals (Netherlands).

Our study was conducted according to the guidelines of the Declaration of Helsinki, priori approval (No. 2/2022) by the ethical committee of Faculty of Pharmacy, Deraya University.

### 2.2 Preparation of isolates for the study

Strains obtained from Hiper® bacterial conjugation kit, Donor strain A was streaked on 2 LB plates with tetracycline (30μg/mL) and the Recipient strain was streaked on LB plates with streptomycin (100μg/mL) and incubated at 37°C overnight.

For clinical strains, Isolation of *E. coli* strains was performed by MacConkey's agar and confirmed biochemically with various screening tests. Disc method was done to select an *E. coli* isolate (Resistant only to tetracycline) and another one (Resistant only to streptomycin) to be used as donor and recipient, respectively. Both genomic and plasmid DNA were extracted from the selected isolates (Qiagen plasmid plus midi kit and Qiaprep spin miniprep kit, USA) and polymerase chain reaction was used to amplify tetracycline and streptomycin resistance

**Table 1. PCR conditions and primer sequences.**

|  | Primer sequences | PCR conditions | Ref. |
|---|---|---|---|
| *traF* | Forward: AAGTGTTCAGGGTGCTTCTGC Reverse: GTCGCCTTAACCGTGGTGTT | Initial denaturation (94ºC for 4 min.), 35 cycles (94ºC for 30 sec., 55ºC for 30 sec., 72ºC for 1 min), 7 min final extension | [11] |
| *tetA* | Forward: GACTATCGTCGCCGCACTTA Reverse: ATAATGGCCTGCTTCTCGCC | Initial denaturation (94ºC for 4 min.), 30 cycles (94ºC for 30 sec., 54ºC for 30 sec., 72ºC for 1 min), 7 min final extension | [11] |
| *strA-strB* | Forward TATCTGCGATTGGACCCTCTG Reverse CATTGCTCATCATTTGATCGGCT | Initial denaturation (95ºC for 60 s), 30 cycles (95ºC for 60 s, 60ºC for 30 s, 72ºC for 60 s). | [24] |

genes and *traF* gene (encoded by conjugative plasmid and required for pilus assembly and plasmid transfer). PCR primer sequences and conditions are listed in Table 1.

## 2.3 Testing antibacterial activity and MIC detection

Minimum inhibitory concentrations (MICs) of tetracycline, Streptomycin and non-antibiotic pharmaceuticals were tested using broth microdilution assay. The 96-well plates were incubated at 37°C for 20 h before the OD600 was measured on the plate reader. Wells containing sterile phosphates buffer saline were used as blank. MICs of the bacterial strains were determined as the concentration of the tested agent which inhibited 90% of the growth. Each bacterial strain under the inhibition of the different agents was tested at least in triplicate Hyoscine butylbromide, metoclopramide HCl and tiemonium methyl sulfate were chosen as the non-antibiotic pharmaceuticals for this study. Concentrations of antibiotics were serially diluted from 8 to 0.03125 μg/ml. Also, non-antibiotic pharmaceuticals were used in therapeutically allowed concentrations [25].

## 2.4 Conjugative transfer in absence and in presence of non-antibiotic pharmaceuticals

**2.4.1 Conjugative transfer in absence of non-antibiotic pharmaceuticals.** A single colony from both donor and recipient strains was inoculated in 6ml of LB broth. Then, incubated at 37°C overnight. One ml of overnight grown culture of standard strains or clinical strains was added to 25 ml LB broth with tetracycline. Three ml of overnight culture of the tested strains were added to 25 ml of LB with streptomycin. Cultures were overnight incubated at 37°C in a shaker. Two hundred μL of each donor and recipient cultures were added to a sterile test tube labeled as conjugated sample after gentle mixing and incubated for 1.5 hr without shaking. Two hundred μL of respective cultures were added to tubes labeled as donor and recipient with gentle mixing and incubated for 1.5hr without shaking. Then, 0.1 ml of each culture was plated on antibiotic containing culture media (one plate containing tetracycline, another plate containing streptomycin and the third containing both tetracycline and streptomycin to isolate transconjugants). Transconjugant colonies' number was divided by donor colonies' number to calculate the conjugative transfer frequency. The value of frequency of conjugal transfer was the average of four different trials. Growth of donor, recipient and transconjugants was not included in calculation due to absence of provided nutrients during the conjugation process [26].

**2.4.2 Conjugative transfer in the presence of non-antibiotic pharmaceuticals.** This work designed a mating model to detect the effect of non-antibiotic pharmaceuticals on conjugative transfer between the tested *E. coli* strains. This was achieved by growing donor and recipient isolates to an OD of 1.8 at 600 nm and mixing donor and recipient in nutrient broth (1:2) ratio in the presence of the following sub-inhibitory concentrations: 0.265 μg/ml (Equivalent to plasma concentration), 0.53 μg/ml, 1.06 μg/ml, 2.12 μg/ml and 4.24 μg/ml for metoclopramide HCl. For hyoscine butyl bromide; 0.005 μg/ml (Equivalent to plasma concentration), 0.01 μg/ml, 0.02 μg/ml, 0.04 μg/ml and 0.08 μg/ml were used. Finally, tiemonium methyl sulfate has sub-inhibitory concentrations of 0.8 μg/ml (plasma concentration), 1.6 μg/ml, 3.2 μg/ml, 6.4 μg/ml and 12.8 μg/ml. Levofloxacin was used at its sub-MIC as a positive control for conjugative transfer (0.125 μg/ml) [27]. After incubation for 3 hr. at 37°C, mixture of donor and recipient (20 μL) was plated on tetracycline-streptomycin nutrient agar plates to count colonies of transconjugants. Transconjugant colonies' number was divided by donor colonies' number to calculate the conjugative transfer frequency. The value of frequency of conjugal transfer was the average of four different trials [11].

## 2.5 Reverse conjugation experiment in the presence of non-antibiotic agents

Transconjugant bacteria were considered as donor cells. Standard and clinical recipients described previously were used for the conjugate mating system. All strains were tested using different concentrations of the tested agents. Then, mixed by vertexing and incubated for 3h at 37˚C without shaking. New transconjugants were used to inoculate agar plates containing antibiotics. Colonies were counted and the frequency of transfer was determined as prescribed above [11].

## 2.6 Verification of conjugative plasmid transfer

**2.6.1 Determination of minimum inhibitory concentration of transconjugants.** Minimum inhibitory concentrations (MICs) of transconjugants against tetracycline and streptomycin were determined by broth microdilution methods. The resulting MICs of transconjugants were compared to donor and recipient cells' MICs [25].

**2.6.2 Detection of plasmids.** Colonies of transconjugants on selective tetracycline-streptomycin agar plates were picked and stored at -80°C in glycerol stock (30%). Qiagen plasmid plus midi kit (Qiagen, USA) was applied for extraction of plasmids from donor and transconjugants. PCR was used to amplify plasmid-encoded *traF* and *tetA* genes in both donor and transconjugant. Agarose gel electrophoresis (1%) was used to observe plasmids and amplicons [26].

## 2.7 Transmission Electron Microscopy (TEM)

Effect of non-antibiotic pharmaceuticals on conjugative activity was studied using TEM (JEM1010, JEOL, Tokyo, Japan). After performing conjugation experiment with non-antibiotic pharmaceuticals and incubation for 24 hr, samples for TEM were collected and imaged. Also, a control sample in absence of non-antibiotic pharmaceuticals was included. Samples were prepared for TEM according to standard guidelines [28].

## 2.8 Statistical analysis

Statistical analysis was performed using GraphPad version 8.3.0 software. All data were obtained from at least three biological replicates and presented as mean ± SD. Unpaired *t*-test

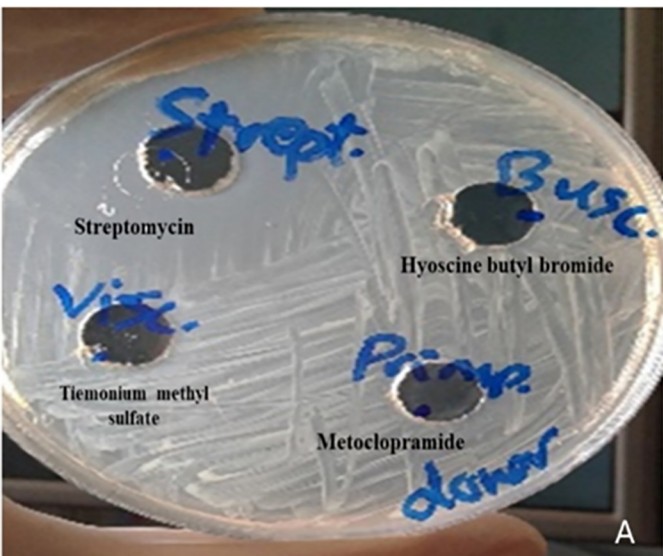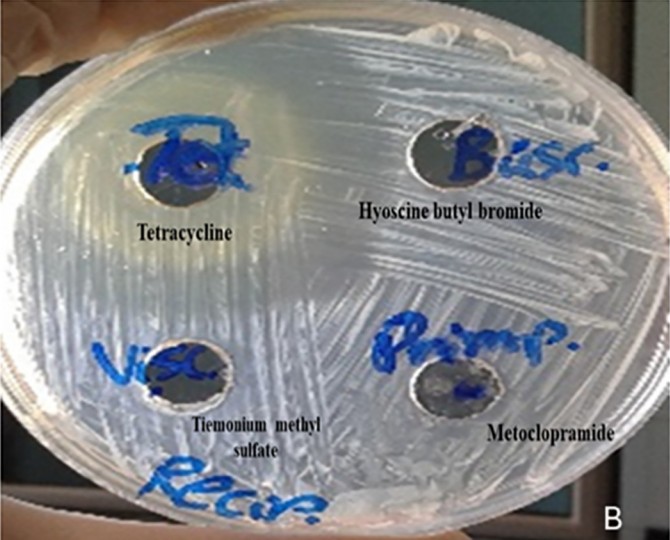

**Fig 1. Antibacterial activity of metoclopramide, hyoscine butyl bromide and tiemonium methyl sulfate: A) against donor isolate in comparison to streptomycin, B) against recipient isolate in comparison to tetracycline.**

(normally distributed data) between two groups or one-way ANOVA among multiple groups were used to calculate *P*-values. Differences with $P < 0.05$ were considered significant [29].

# 3. Results

## 3.1 Antibacterial activity of non-antibiotic pharmaceuticals

Broth microdilution assay illustrated no inhibitory activity for metoclopramide, hyoscine butyl bromide and tiemonium methyl sulfate (MIC > 1024 μg/ml). Agar well diffusion assay was used to confirm the previous findings by reporting no inhibitory activity for the non-antibiotic pharmaceuticals against donor and recipient isolates (Fig 1).

## 3.2 Non-antibiotic pharmaceuticals have different effects against ARGs conjugative transfer

For both standard and clinical *E. coli* strains; non-antibiotic pharmaceuticals were added at therapeutic concentrations. Metoclopramide HCl, at all five concentrations (from 0.265 to 4.24 μg/ml), the number of transconjugants dramatically increased ($P < 0.05$). Hyoscine butyl bromide, at concentrations from 0.005 to 0.01 μg/ml, slightly increased transconjugant number ($P < 0.05$). In contrast, tiemonium methyl sulfate at concentrations from 0.8 to 12.8 μg/ml, decreased the transconjugant number ($P < 0.05$). A threshold effect was observed for the pharmaceuticals under study, as the gradual increasing of the dose concentration influenced the conjugation response.

Fig 2a illustrates transconjugant colonies in absence of antibiotics and non-antibiotic pharmaceuticals as a control. Efficiency of conjugation between donor and recipient isolates was greatly promoted by metoclopramide HCl (Fig 2b) while was dramatically decreased by hyoscine butylbromide (Fig 2c). Also, slight induction of conjugal transfer by tiemonium methylsulfate was observed (Fig 2d). Levofloxacin at sub-MIC level greatly increased conjugal transfer compared to control conjugation tube (Fig 2e). Transfer ratio (expressed as number of transconjugants per recipient cells) and fold change of transfer ratio were illustrated for the

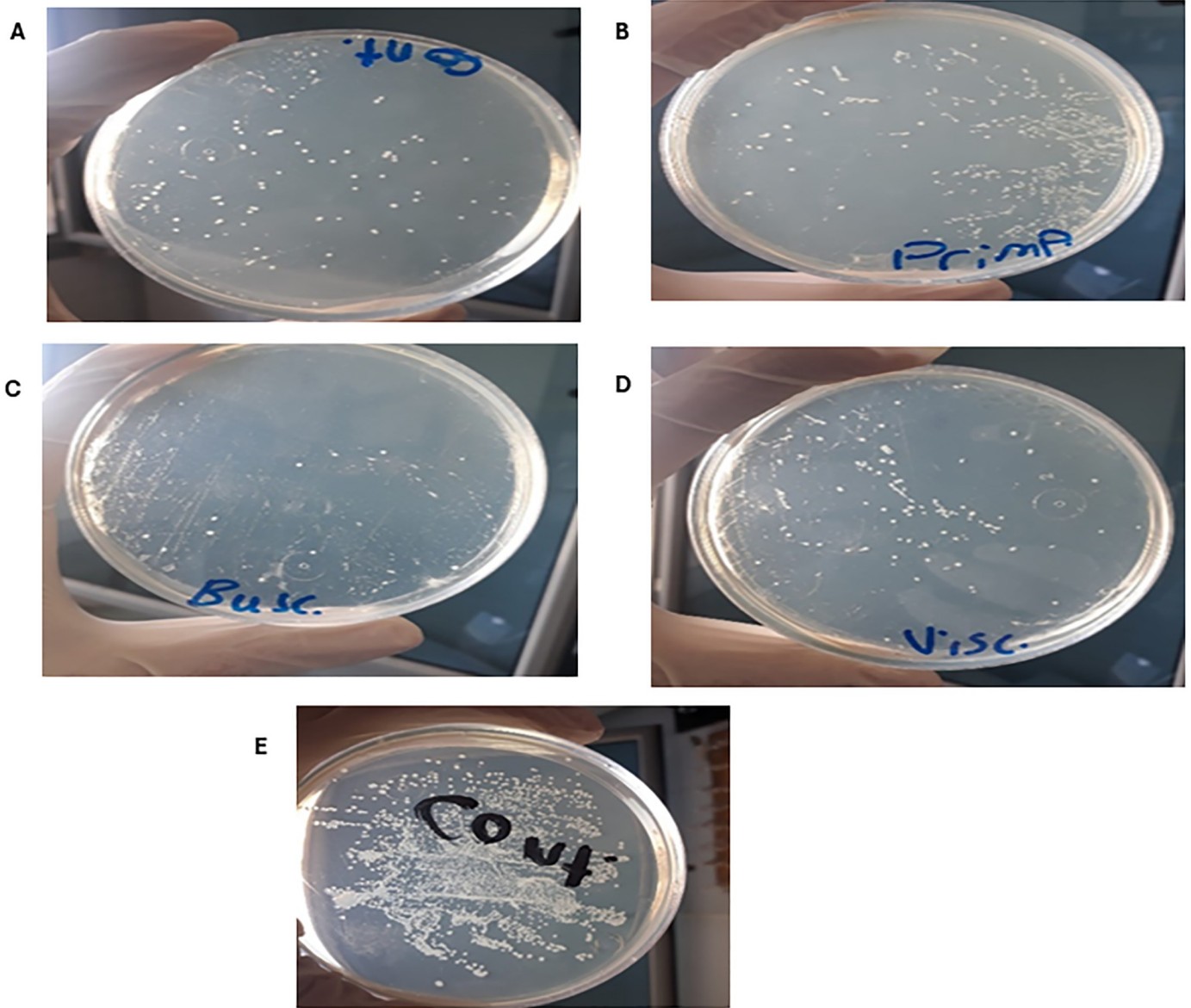

**Fig 2. Effect of A) Control, B) Metoclopramide HCl, C) Hyoscine butylbromide, D) Tiemonium methylsulfate, and E) Levofloxacin at sub-MIC on conjugal transfer frequency.**

tested non-antibiotic agents at different concentrations and levofloxacin at sub-MIC (Table 2). Fold change of transfer ratio was greatly enhanced in presence of metoclopramide HCl (Fig 3) and slightly increased with tiemonium methylsulfate (Fig 4). However, it was decreased in presence of hyoscine butylbromide as compared to negative control (Fig 5).

### 3.3 Reverse conjugation experiment in the presence of non-antibiotic agents

To test if the transconjugant is transferable, the reverse conjugative experiment is performed. The plasmid is transferred from conjugants to donners. It was seen that the conjugative

**Table 2. Conjugative transfer under the exposure of different concentrations of non-antibiotic pharmaceuticals.**

| | Number of recipients in 1 ml | Absolute number of transconjugants in 1 ml | Fold change of transconjugant absolute number | Transfer ratio (number of transconjugant / number of recipient) | Fold change of Transfer ratio |
|---|---|---|---|---|---|
| Control | $2.5 \times 10^4 \pm 817$ | $\pm 745.72$ | $1.00 \pm 0.08$ | $29.7 \times 10^{-4} \pm 3 \times 10^{-4}$ | $1.00 \pm 0.11$ |
| **Metoclopramide HCl** | | | | | |
| 0.265 µg/ml | $2.3 \times 10^4 \pm 12415$ | $485 \pm 8.16$ | $6.55 \pm 0.4$ | $195 \times 10^{-4} \pm 16 \times 10^{-4}$ | $6.55 \pm 0.16^*$ |
| 0.53 µg/ml | $2.4 \times 10^4 \pm 12034$ | $523 \pm 16.33$ | $7.06 \pm 0.33$ | $238 \times 10^{-4} \pm 31 \times 10^{-4}$ | $8.2 \pm 1.5^*$ |
| 1.06 µg/ml | $2.2 \times 10^4 \pm 11002$ | $578 \pm 81.65$ | $7.81 \pm 0.51$ | $240 \times 10^{-4} \pm 23 \times 10^{-4}$ | $7.97 \pm 0.2^*$ |
| 2.12 µg/ml | $2.5 \times 10^4 \pm 12472$ | $634 \pm 27.76$ | $8.57 \pm 1.05$ | $255 \times 10^{-4} \pm 14 \times 10^{-4}$ | $8.67 \pm 0.5^*$ |
| 4.24 µg/ml | $2.4 \times 10^4 \pm 11885$ | $670 \pm 81.65$ | $9.05 \pm 1.82$ | $280 \times 10^{-4} \pm 41 \times 10^{-4}$ | $9.73 \pm 2.4^*$ |
| **Hyoscine butylbromide** | | | | | |
| 0.005 µg/ml | $2.5 \times 10^4 \pm 817$ | $39 \pm 16.33$ | $0.53 \pm 0.18$ | $16 \times 10^{-4} \pm 7 \times 10^{-4}$ | $0.51 \pm 0.18^*$ |
| 0.01 µg/ml | $2.5 \times 10^4 \pm 4082$ | $32 \pm 8.16$ | $0.43 \pm 0.08$ | $14 \times 10^{-4} \pm 6 \times 10^{-4}$ | $0.44 \pm 0.14^*$ |
| 0.02 µg/ml | $2.4 \times 10^4 \pm 816$ | $28 \pm 12.25$ | $0.38 \pm 0.2$ | $12 \times 10^{-4} \pm 5 \times 10^{-4}$ | $0.42 \pm 0.23^*$ |
| 0.04 µg/ml | $2.2 \times 10^4 \pm 3266$ | $19 \pm 4.08$ | $0.26 \pm 0.08$ | $9 \times 10^{-4} \pm 3 \times 10^{-4}$ | $0.32 \pm 0.14^*$ |
| 0.08 µg/ml | $2.4 \times 10^4 \pm 4082$ | $\pm 155.72$ | $0.20 \pm 0.06$ | $7 \times 10^{-4} \pm 3 \times 10^{-4}$ | $0.22 \pm 0.08^*$ |
| **Tiemonium methylsulfate** | | | | | |
| 0.8 µg/ml | $2.5 \times 10^4 \pm 2449$ | $74 \pm 8.16$ | $1.00 \pm 0.19$ | $29.6 \times 10^{-4} \pm 4 \times 10^{-4}$ | $1.01 \pm 0.12$ |
| 1.6 µg/ml | $2.5 \times 10^4 \pm 3266$ | $78 \pm 14.7$ | $1.05 \pm 0.28$ | $31 \times 10^{-4} \pm 2 \times 10^{-4}$ | $1.06 \pm 0.18$ |
| 3.2 µg/ml | $2.3 \times 10^4 \pm 5312$ | $83 \pm 10.6$ | $1.12 \pm 0.23$ | $37 \times 10^{-4} \pm 8 \times 10^{-4}$ | $1.26 \pm 0.3$ |
| 6.4 µg/ml | $2.5 \times 10^4 \pm 4899$ | $87 \pm 5.72$ | $1.18 \pm 0.17$ | $37 \times 10^{-4} \pm 5 \times 10^{-4}$ | $1.24 \pm 0.05$ |
| 12.8 µg/ml | $2.4 \times 10^4 \pm 6532$ | $88 \pm 13.06$ | $1.19 \pm 0.27$ | $40 \times 10^{-4} \pm 17 \times 10^{-4}$ | $1.42 \pm 0.74$ |
| **Levofloxacin at sub-MIC (0.125 µg/ml)** | $2.2 \times 10^4 \pm 1633$ | $2500 \pm 408$ | $33.78 \pm 8.21$ | $1150 \times 10^{-4} \pm 253 \times 10^{-4}$ | $40.08 \pm 13.2^*$ |

Results are shown as mean ± SD, significant differences between non-antibiotic dosed samples and the control were analyzed by independent-sample *t* test.

\* *P* < 0.05 were considered significant.

transfer frequency was enhanced in the presence of metoclopramide HCl as compared to negative control. Also, an increase in the fold changes of the transfer frequency occurred in the presence of the tested drug. Transconjugant colonies were chosen and re-grown on selection plates to verify the identity of the recipient cells.

## 3.4 Determination of minimum inhibitory concentration of transconjugants

The MICs of the transconjugants against tetracycline and streptomycin were detected (Table 3) and were the same or higher than the donor or recipients MICs.

## 3.5 Detection of plasmids

Confirmation of successful conjugal transfer was done by plasmid analysis on agarose gel electrophoresis. Conjugative plasmid was detected in donor and transconjugant strains as illustrated by first and second lanes (Fig 6). Recipient isolate did not show any plasmid as recorded in lane 3.

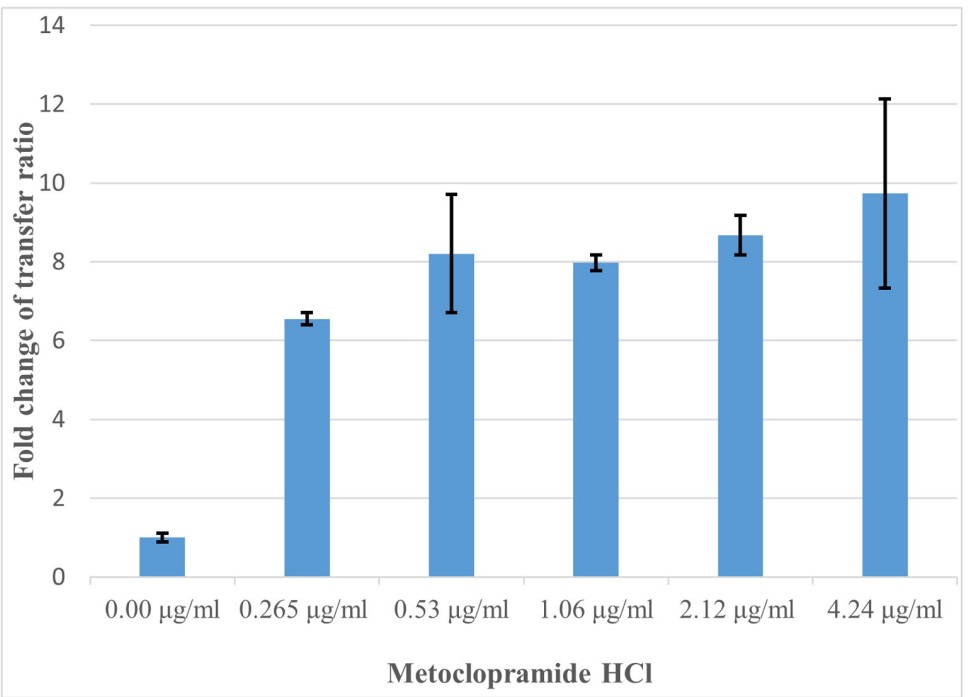

**Fig 3. Fold changes of transfer ratio under the exposure of metoclopramide HCl at different concentrations.**

## 3.6 Effect of non-antibiotic pharmaceuticals on conjugation efficiency

Fig 4 illustrates TEM images of conjugative transfer in the presence of non-antibiotic pharmaceuticals and in their absence as a control (Fig 7A1 and 7A2). Metoclopramide HCl induced conjugal transfer between donor and recipient as seen in the close contact between cells,

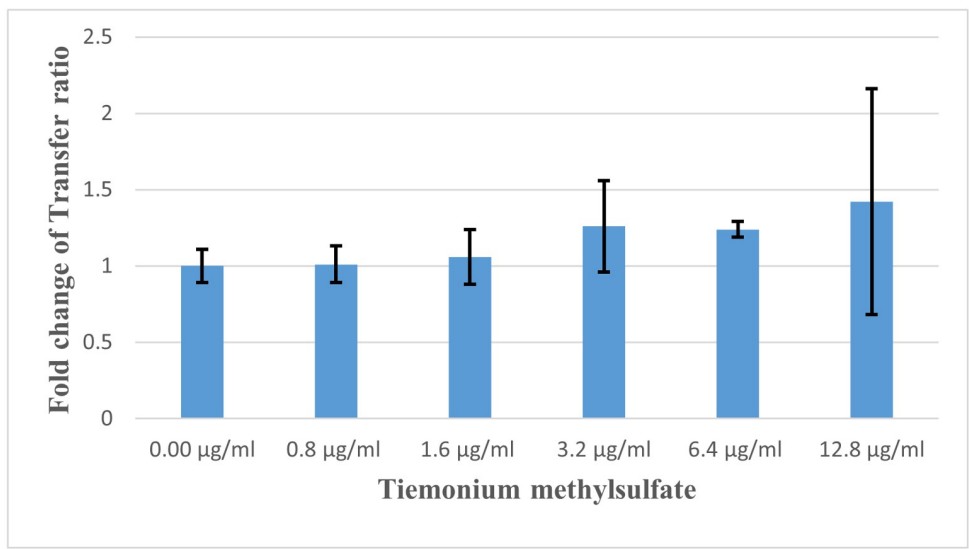

**Fig 4. Fold changes of transfer ratio under the exposure of tiemonium methyl sulfate at different concentrations.**

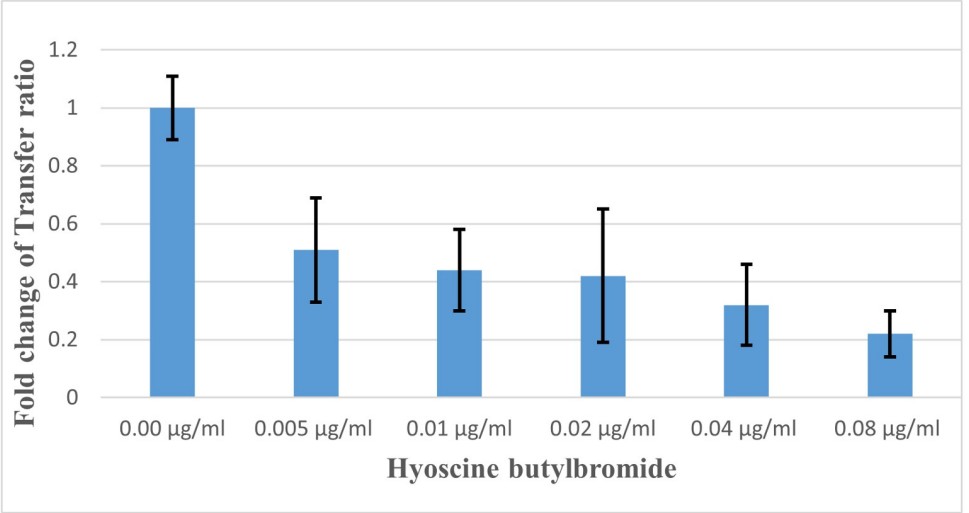

**Fig 5. Fold changes of transfer ratio under the exposure of hyoscine butyl bromide at different concentrations.**

periplasmic bridges and aggregation of cells (Fig 7B1 and 7B2). Hyoscine butylbromide decreased conjugal activity as observed in the little affinity between cells (Fig 7C).

## 4. Discussion

The present findings highlight the effect of non-antibiotic pharmaceuticals on bacterial behavior and how this will influence the dissemination of antibiotic resistance between isolates changing the duration of activity of used antibiotics. Antimicrobial resistance is considered a major health problem which arose mainly due to the misuse of antibiotics all over the world. This dilemma is usually attributed to various causes such as: availability of antibiotics everywhere and health care persons may suffer from lack of education. Also, many patients stop treatment with antibiotics upon feeling better, use leftover antibiotics and may use antibiotics in treating viral infection [30–33].

The rapid dissemination of ARGs among bacterial species occurs through HGT. Conjugation is a common mechanism by which bacteria horizontally transfer their resistance genes for adaptation with stress conditions such as continuous antibiotic exposure and lack of nutrients [34]. Bacterial conjugation is a major promoter for bacterial genome evolution which supply bacteria by required genes for antimicrobial resistance, biofilm formation, virulence and heavy metal resistance [35, 36]. Sub-inhibitory concentration of antibiotics has been found to induce dissemination of antibiotic resistance by HGT particularly through conjugation and transformation [11, 37].

**Table 3. Minimum inhibitory concentrations (MICs) of donor, recipient, and different transconjugants towards antibiotics.**

| Antibiotics | MICs (mg/L) | | | | |
|---|---|---|---|---|---|
| | Donor | Recipient | TC 1 | TC 2 | TC 3 |
| Tetracycline | 32 | 4 | 32 | 32 | 32 |
| Streptomycin | 4 | 64 | 64 | 64 | 64 |

*TC 1–3: transconjugants in mating system treated with hyoscine butylbromide, metoclopramide HCl and tiemonium methyl sulfate, respectively.

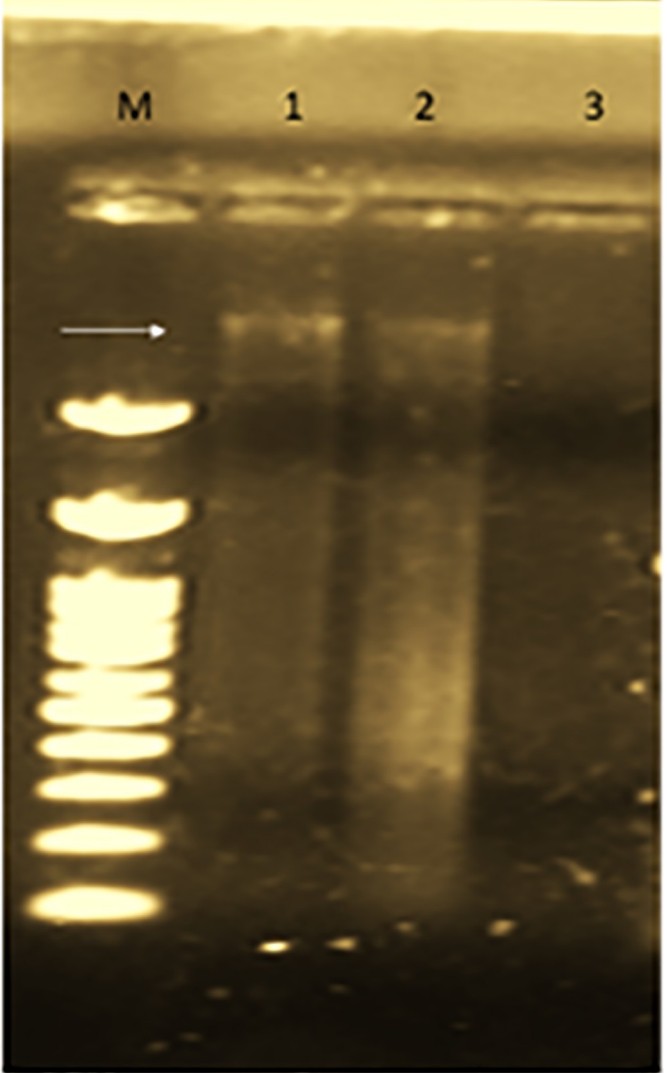

**Fig 6. Plasmid detection in donor (lane 1), transconjugants (lane 2) and recipient (lane 3).**

Not only antibiotics affect gene transfer between bacteria but also non-antibiotic pharmaceuticals may be supposed to have an effect as recorded for more than 200 non-antibiotic pharmaceuticals which influence gut bacteria like antibiotics do [15]. All over the world, non-antibiotic pharmaceuticals represent 95% of the total drugs in the market. Although their wide distribution, little is known about how non-antibiotic pharmaceuticals affect HGT between bacteria [38, 39].

Antiemetic and antispasmodic drugs are widely used by patients during their course of treatment of GIT infection caused by *E. coli*. So, in the present work, effect of metoclopramide HCl, hyoscine butylbromide and tiemonium methylsulfate on conjugal transfer between *E. coli* isolates was studied. Also, levofloxacin at sub-inihibitory concentration was included. Hyoscine butylbromide and tiemonium methylsulfate have antispasmodic activity while metoclopramide HCl has antiemetic effect. Metoclopramide HCl has antibiotic-like effect behavior as for promoting conjugal transfer between isolates especially when antibiotic is administered

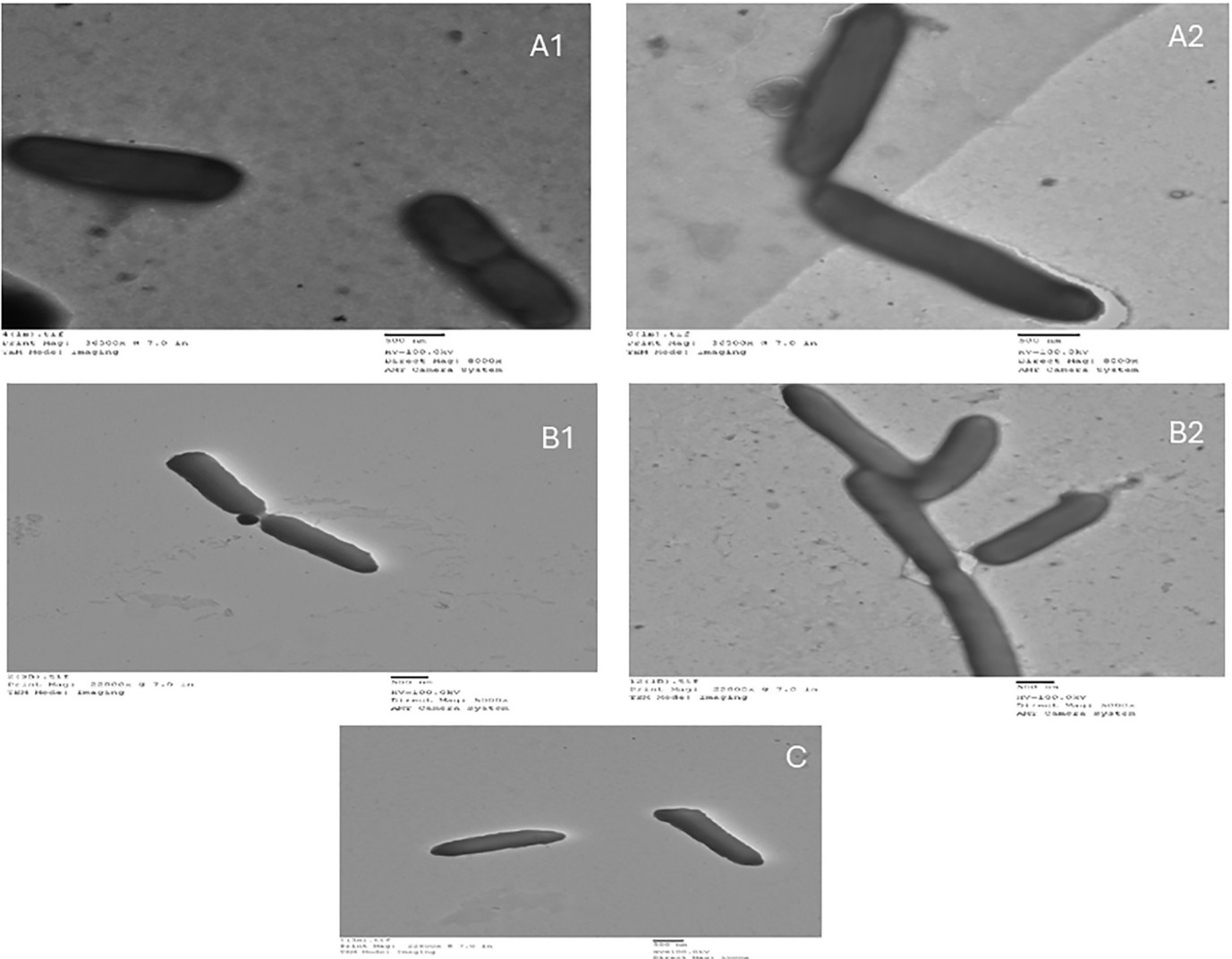

**Fig 7. TEM images for conjugal activity between donor and recipient A) Control, B) In presence of metoclopramide HCl, C) In presence of hyoscine butyl bromide.**

at sub-inhibitory concentration. While conjugal transfer frequency is dramatically increased by metoclopramide HCl, it was obviously suppressed by hyoscine butylbromide. This means that co-administration of metoclopramide HCl as antiemetic with antibiotics may promote the transfer of antibiotic resistance genes among pathogenic bacteria in the GIT decreasing the susceptibility to the antibiotic in the long run use.

Acceleration of dissemination of antibiotic resistance was reported for non-antibiotic pharmaceuticals such as ibuprofen, naproxen, diclofenac, gemfiprozil and propranolol as plasmid-borne conjugal transfer was promoted between bacteria of the same and different genera [11]. Also, dissemination of plasmid-mediated antibiotic resistance was induced by the antiepileptic carbamazepine [40]. These findings agree with what was obtained in our study for metoclopramide HCl. Norepinephrine has not revealed any effect on plasmid conjugal transfer between *E. coli* isolates [41]. The previous result agrees with what achieved in the present work for

tiemonium methylsulfate which caused a slight increase in conjugal transfer between *E. coli* isolates [41]. In the present study, hyoscine butylbromide dramatically decreased conjugal transfer between tested *E. coli* supporting the activity of co-administered antibiotics by weakening the resistance of targeted pathogenic bacteria. Expired drug metoclopramide HCl indicated antibacterial activity for carbon steel in 0.5 M H3PO4 solution [42] which disagrees with what achieved in the present work as none of the tested non-antibiotic pharmaceuticals revealed antibacterial activity against isolates.

TEM imaging was applied to reveal changes of cell arrangement as an indicator to study the effect of metoclopramide HCl and hyoscine butylbromide on bacterial cells. Aggregation of cells, formation of periplasmic bridges and decreasing distances between cells were obviously noticed with metoclopramide HCl while with hyoscine butylbromide little affinity between cells was observed. In absence of non-antibiotic pharmaceuticals, cells were separated and intact but in presence of non-antibiotic pharmaceuticals, cells became closer to each other with partially damaged cell membrane [11].

In the present findings, the role of non-antibiotic pharmaceuticals in dissemination of antibiotic resistance between pathogenic gastrointestinal tract *E. coli* isolates was clearly demonstrated. Previous findings turn on alarm for bacterial response towards different administered pharmaceuticals highlighting the bacterial behavior when non-antibiotic pharmaceuticals are co-administered with antibiotics. As antibiotics at sub-inhibitory concentrations play important role in disseminating resistance genes through conjugation; it is suggested that some non-antibiotic pharmaceuticals like antiemetic drugs play the same role. Increased ROS production and cell membrane permeability were positively correlated with enhanced conjugation. Furthermore, these non-antibiotic drugs produced reactions like improved efflux pumps that is seen when bacteria are exposed to antibiotics. More work is required to illustrate the effect of pharmaceuticals in the market on the transfer of resistance genes and how this would contribute to maximize or minimize the antibiotic resistance crisis.

## 5. Conclusion

In prescribing antibiotics for treating intestinal pathogenic *E.coli*, antiemetic metoclopramide HCl was found to increase the spread of resistance against the co-administered antibiotic by promoting the plasmid-mediated conjugal transfer of antibiotic resistance genes leading to increasing the duration of infection, followed by tiemonium methylsulfate. While, antispasmodic hyoscine butylbromide showed lower rate of conjugal transfer in comparison to control.

## Supporting information

**S1 File.**
(DOCX)

**S2 File.**
(XLSX)

## Author Contributions

**Conceptualization:** Eman Farouk Ahmed.

**Data curation:** Doaa Safwat Mohamed, Rehab Mahmoud Abd El-Baky, Sahar K. Ghanem, Ramadan Yahia, Alaa M. Alqahtani.

**Formal analysis:** Sahar K. Ghanem, Ramadan Yahia.

**Funding acquisition:** Alaa M. Alqahtani, Mohammed A. S. Abourehab.

**Investigation:** Mohamed Ahmed El-Mokhtar.

**Methodology:** Doaa Safwat Mohamed, Alaa M. Alqahtani, Mohammed A. S. Abourehab, Eman Farouk Ahmed.

**Supervision:** Rehab Mahmoud Abd El-Baky, Ramadan Yahia, Mohammed A. S. Abourehab.

**Validation:** Sahar K. Ghanem.

**Visualization:** Eman Farouk Ahmed.

**Writing – original draft:** Doaa Safwat Mohamed, Rehab Mahmoud Abd El-Baky, Mohamed Ahmed El-Mokhtar, Sahar K. Ghanem, Ramadan Yahia, Eman Farouk Ahmed.

**Writing – review & editing:** Rehab Mahmoud Abd El-Baky, Alaa M. Alqahtani, Mohammed A. S. Abourehab.

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
