## [Decision Letter · Decision Letter 0]

14 Mar 2024

PONE-D-23-09629

Role of Some Antiemetics and Antispasmodics in the dissemination of antibiotic resistance genes among intestinal pathogenic E. coli

PLOS ONE

Dear Dr. Abd El-Baky,

Thank you for submitting your manuscript to PLOS ONE. After careful consideration, we feel that it has merit but does not fully meet PLOS ONE’s publication criteria as it currently stands. Therefore, we invite you to submit a revised version of the manuscript that addresses the points raised during the review process.

I would like to sincerely apologise for the delay you have incurred with your submission. It has been exceptionally difficult to secure reviewers to evaluate your study. We have now received four completed reviews; the comments are available below. The reviewers have raised significant scientific concerns about the study that need to be addressed in a revision.

Please revise the manuscript to address all the reviewer's comments in a point-by-point response in order to ensure it is meeting the journal's publication criteria. Please note that the revised manuscript will need to undergo further review, we thus cannot at this point anticipate the outcome of the evaluation process.

One or more of the reviewers has recommended that you cite specific previously published works. Members of the editorial team have determined that the works referenced are not directly related to the submitted manuscript. As such, please note that it is not necessary or expected to cite the works requested by the reviewer. 

We look forward to receiving your revised manuscript.

Kind regards,

Miquel Vall-llosera Camps

Staff Editor

PLOS ONE

Journal Requirements:

Reviewers' comments:

Reviewer's Responses to Questions

**Comments to the Author**

1. Is the manuscript technically sound, and do the data support the conclusions?

Reviewer #1: Yes

Reviewer #2: Yes

Reviewer #3: Yes

Reviewer #4: Partly

2. Has the statistical analysis been performed appropriately and rigorously? 

Reviewer #1: Yes

Reviewer #2: Yes

Reviewer #3: Yes

Reviewer #4: I Don't Know

3. Have the authors made all data underlying the findings in their manuscript fully available?

Reviewer #1: Yes

Reviewer #2: Yes

Reviewer #3: Yes

Reviewer #4: Yes

4. Is the manuscript presented in an intelligible fashion and written in standard English?

Reviewer #1: Yes

Reviewer #2: Yes

Reviewer #3: Yes

Reviewer #4: No

5. Review Comments to the Author

Reviewer #1: Dear authors, You are doing well

Please track the requested revisions on the attached file. Your good tracking and revising all the comments to create the manuscript is so suitable for the publication

Reviewer #2: The article is well written and scientifically correct, I have just 2 minor comments

1- Write, ‘..to be used as donor and recipient, respectively.’

2- First paragraph of discussion, write “..The rapid dissemination of ARGs among bacterial species occurs through HGT.’

Reviewer #3: General:

• The manuscript investigates the role of certain antiemetics and antispasmodics in disseminating antibiotic-resistance genes among pathogenic E. coli in the gastrointestinal tract. It presents an important study on the effect of non-antibiotic pharmaceuticals on the conjugal transfer of resistance genes.

• No line numbers and it wasn't easy to review.

• Maybe authors could use the word antimicrobial instead of antibiotics to improve breadth and context.

• The manuscript could be further improved by improving English language usage.

• It would have been great if the authors had explained the selection criteria for the non-antibiotic pharmaceuticals studied and their relevance to antibiotic resistance.

Specific issues

Abstract:

1. Check all abbreviations have a full form. For example, the word GIT is written only in abbreviated form.

Introduction:

• The meaning of the sentence:... Surprisingly, it was recently documented that the consumption of non-antibiotic prescribed drugs represents about ninety fifth of the drug market [14],…." is confusing.

Please use more technical words like detection rate, prevalent, rather than ….." are highly detected [18]."

• The existing knowledge on the role of antiemetics and antispasmodics on antimicrobial resistance development and how the current study is going to fill the existing knowledge gap isn't clear.

Discussion:

• More emphasis is given to antibiotics rather than antiemetics and antispasmodics, especially in the first paragraph.

• Some sentences lack citation. For example: "The previous result agrees with what was achieved in the present work for tiemonium methylsulfate, which caused a slight increase in conjugal transfer between E. coli isolates."

• I recommend further elaborating on the mechanisms by which metoclopramide HCl enhances the conjugal transfer of resistance genes and the implications of these findings in a clinical setting.

Reviewer #4: 1. Which standard protocols were used by authors for MIC?

2. English editing is highly recommended by a native speaker. Some part is difficult to understand.

3. All figure legends must be clear and informative.

4. Table 1 is not necessary. Put it in Suppl. Files

5. In figure 1: Which culture method used?

6. Figure 6: the plasmid band is vague. replace with better fig.

6. Figure 7: The quality is poor. the bar should be added for TEM.

6. PLOS authors have the option to publish the peer review history of their article (what does this mean?). If published, this will include your full peer review and any attached files.

Reviewer #1: No

Reviewer #2: **Yes: **Mohamed Salah Abbassi

Reviewer #3: No

Reviewer #4: **Yes: **Abdollah Derakhshadeh

---

## [Author Response · Author response to Decision Letter 0]

3 Apr 2024

Response to reviewer was attached as word file

---

## [Decision Letter · Decision Letter 1]

29 Apr 2024

PONE-D-23-09629R1Role of Some Antiemetics and Antispasmodics in the dissemination of antibiotic resistance genes among intestinal pathogenic Escherichia coliPLOS ONE

Dear Dr. Abd El-Baky,

Thank you for submitting your manuscript to PLOS ONE. After careful consideration, we feel that it has merit but does not fully meet PLOS ONE’s publication criteria as it currently stands. Therefore, we invite you to submit a revised version of the manuscript that addresses the points raised during the review process.

We look forward to receiving your revised manuscript.

Kind regards,

Farah Al-Marzooq, MD, PhD

Academic Editor

PLOS ONE

Journal Requirements:

Reviewers' comments:

Reviewer's Responses to Questions

**Comments to the Author**

1. If the authors have adequately addressed your comments raised in a previous round of review and you feel that this manuscript is now acceptable for publication, you may indicate that here to bypass the “Comments to the Author” section, enter your conflict of interest statement in the “Confidential to Editor” section, and submit your "Accept" recommendation.

Reviewer #1: All comments have been addressed

Reviewer #3: (No Response)

2. Is the manuscript technically sound, and do the data support the conclusions?

Reviewer #1: Yes

Reviewer #3: Partly

3. Has the statistical analysis been performed appropriately and rigorously? 

Reviewer #1: Yes

Reviewer #3: N/A

4. Have the authors made all data underlying the findings in their manuscript fully available?

Reviewer #1: Yes

Reviewer #3: Yes

5. Is the manuscript presented in an intelligible fashion and written in standard English?

Reviewer #1: Yes

Reviewer #3: No

6. Review Comments to the Author

Reviewer #1: Dear authors. You are doing well. The manuscript is worthy publication. Hope the scientific progression

Reviewer #3: General comments

• A thorough proofreading by a native English speaker or professional editor is recommended to refine the language. A few typographical errors and grammatical issues throughout the text need correction. For instance, consistency in the use of terminology such as "non-antibiotics" versus "non-antibiotic pharmaceuticals" should be addressed. The words antibiotic and antimicrobial are both used interchangeably in the manuscript. Use one term/word consistently to explain one thing. Some sentences are overly complex and could be simplified for better readability.

Title and abstract

• Title: The current title could be revised for improvement. A proposed revision might be: "Influence of Selected Non-Antibiotic Pharmaceuticals on Antibiotic Resistance Gene Transfer in Escherichia coli."

Introduction

• The introduction outlines the background and the significance of the research. It would enhance the manuscript to include a brief explanation of previous studies directly linking non-antibiotic pharmaceuticals with conjugative transfer, providing a stronger justification for the current study.

• Line 80: the words prevalent and rate are the same here. Use detection rate or prevalent.

Materials and methods

• Line 109: what does “Raw materials” refer to?

• Line 109-113: in some instances, product numbers are indicated in parenthesis, while in others, the country where the product is made is indicated. Please revise and follow the journal guidelines.

Results

• Was there any dose-based response? If applicable, describe the dose-response relationships and whether a threshold effect is observed for the pharmaceuticals under study.

Discussion

• The entire discussion is written as one paragraph, making it difficult to follow the points. Please revise the discussion part by making each main point a paragraph.

• The discussion effectively ties the results back to the broader implications of antibiotic resistance. Expanding on the molecular mechanisms potentially involved in the observed effects could enrich the discussion. For example, are there known interactions at the cellular or molecular level between the pharmaceuticals studied and bacterial cell membranes or DNA transfer machinery?

7. PLOS authors have the option to publish the peer review history of their article (what does this mean?). If published, this will include your full peer review and any attached files.

Reviewer #1: **Yes: **Mushtak T.S.Al-Ouqaili

Reviewer #3: No

---

## [Author Response · Author response to Decision Letter 1]

14 May 2024

General comments: A thorough proofreading by a native English speaker or professional editor is recommended to refine the language. A few typographical errors and grammatical issues throughout the text need correction. For instance, consistency in the use of terminology such as "non-antibiotics" versus "non-antibiotic pharmaceuticals" should be addressed. The words antibiotic and antimicrobial are both used interchangeably in the manuscript. Use one term/word consistently to explain one thing. Some sentences are overly complex and could be simplified for better readability. "non-antibiotics" versus "non-antibiotic pharmaceuticals" is addressed. One term/word is used consistently to explain one thing

Title and abstract

Title: The current title could be revised for improvement. A proposed revision might be: "Influence of Selected Non-Antibiotic Pharmaceuticals on Antibiotic Resistance Gene Transfer in Escherichia coli." Done. We changed the title to the proposed one

Introduction

The introduction outlines the background and the significance of the research. It would enhance the manuscript to include a brief explanation of previous studies directly linking non-antibiotic pharmaceuticals with conjugative transfer, providing a stronger justification for the current study. Done

Line 80: the words prevalent and rate are the same here. Use detection rate or prevalent. Done

Materials and methods

Line 109: what does “Raw materials” refer to? Refer to the used antibiotics. It is deleted

Line 109-113: in some instances, product numbers are indicated in parenthesis, while in others, the country where the product is made is indicated. Please revise and follow the journal guidelines. Done

Results

Was there any dose-based response? If applicable, describe the dose-response relationships and whether a threshold effect is observed for the pharmaceuticals under study. Yes and it is described in the result section

Discussion

The entire discussion is written as one paragraph, making it difficult to follow the points. Please revise the discussion part by making each main point a paragraph. Done

The discussion effectively ties the results back to the broader implications of antibiotic resistance. Expanding on the molecular mechanisms potentially involved in the observed effects could enrich the discussion. For example, are there known interactions at the cellular or molecular level between the pharmaceuticals studied and bacterial cell membranes or DNA transfer machinery? Yes and the interactions at the cellular or molecular level between the pharmaceuticals and DNA transfer is discussed in the Discussion section

---

## [Editor Report · Decision Letter 2]

22 May 2024

Influence of Selected Non-Antibiotic Pharmaceuticals on Antibiotic Resistance Gene Transfer in Escherichia coli

PONE-D-23-09629R2

Dear Dr. Abd El-Baky,

We’re pleased to inform you that your manuscript has been judged scientifically suitable for publication and will be formally accepted for publication once it meets all outstanding technical requirements.

Kind regards,

Farah Al-Marzooq, MD, PhD

Academic Editor

PLOS ONE
---

## [Editor Report · Acceptance letter]

30 May 2024

PONE-D-23-09629R2 

PLOS ONE

Dear Dr. Abd El-Baky, 

I'm pleased to inform you that your manuscript has been deemed suitable for publication in PLOS ONE. Congratulations! Your manuscript is now being handed over to our production team.

Kind regards, 

on behalf of

Dr. Farah Al-Marzooq 

Academic Editor

PLOS ONE